# Reducing the Covariate Shift by Mirror Samples in Cross Domain Alignment

**Yin Zhao**[*†]
Alibaba Group
yinzhao.zy@alibaba-inc.com

**Minquan Wang**[*]
Alibaba Group
minquan.wmq@alibaba-inc.com

**Longjun Cai**
Alibaba Group
longjun.clj@alibaba-inc.com

## Abstract

Eliminating the covariate shift cross domains is one of the common methods to deal with the issue of domain shift in visual unsupervised domain adaptation. However, current alignment methods, especially the prototype based or sample-level based methods neglect the structural properties of the underlying distribution and even break the condition of covariate shift. To relieve the limitations and conflicts, we introduce a novel concept named (virtual) mirror, which represents the equivalent sample in another domain. The equivalent sample pairs, named mirror pairs reflect the natural correspondence of the empirical distributions. Then a mirror loss, which aligns the mirror pairs cross domains, is constructed to enhance the alignment of the domains. The proposed method does not distort the internal structure of the underlying distribution. We also provide theoretical proof that the mirror samples and mirror loss have better asymptotic properties in reducing the domain shift. By applying the virtual mirror and mirror loss to the generic unsupervised domain adaptation model, we achieved consistently superior performance on several mainstream benchmarks.

## 1   Introduction

Current deep learning models have achieved significant progress on many tasks but heavily rely on the large amount of labeled data. The generality of the model may be severely degraded when facing the same task of a different domain. So, domain adaptation (DA) attracts a lot of attentions in recent years. DA has several different settings, such as unsupervised [34] or semi-supervised DA [23], open-set [6] or closed-set DA [38], as well as single or multi-source DA [40]. In this paper, we consider the closed-set, single-source unsupervised domain adaptation (UDA) on the classification task. In this setting, one has the source domain data with labels and is expected to predict for the unlabeled target domain data. Both the source and target domains share the same class labels.

The main challenge for DA is the domain shift. It can be further categorized into covariate shift [45, 44], target/label shift [48]),etc. Specifically, define $p^s(x, y)$ and $p^t(x, y)$ as the joint distributions of the source and target domains. The covariate shift refers to the difference of the marginal distribution of $x$, i.e. $p^s(x) \neq p^t(x)$, assuming the conditional probabilities cross domains are same, i.e. $p^s(y|x) = p^t(y|x)$. Most of the current methods, in terms of learning domain-invariant representation or domain alignment, are working to reduce the covariate shift explicitly or implicitly.

---

[*]Equal Contribution
[†]Corresponding Author

35th Conference on Neural Information Processing Systems (NeurIPS 2021).

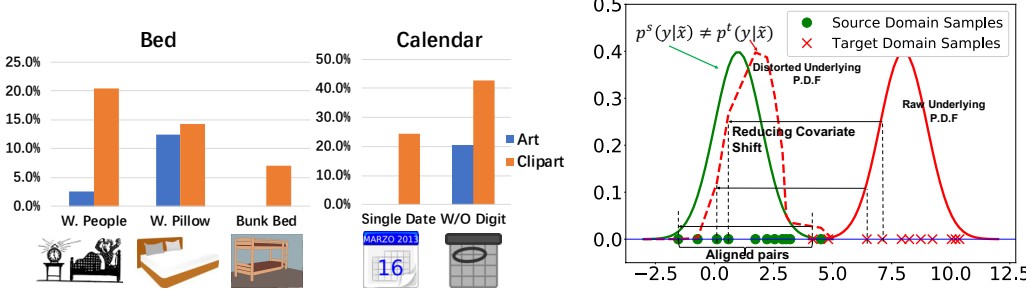

(a) Example of visual patterns and ratios.

(b) One-dimension illustration of the dilemma.

Figure 1: (a) The ratios of samples having certain visual patterns for the same class (Bed and Calendar) from different domains (Art v.s. Clipart) in Office-Home. (b) In one-dimension case, aligning the biased datasets of different domains (green dots vs red cross) will distort the underlying distribution (red dashed line), making the conditional probability in the alingned space ($\tilde{x}$) not same, i.e. $p^s(y|\tilde{x}) \neq p^t(y|\tilde{x})$

They aligned the domain centers [34, 8, 4, 47] or class-wise centers/prototypes [37], regularized the margins of inter-intra class distances [7, 15, 28] and even aligned the two domains through sample-level mappings[16, 14, 13].

However, there exists a covert dilemma between the covariate shift and its assumption in practice, which bounds the performance of those methods. That is reducing the covariate shift will break its assumptions empirically. The underlying reason is that the dataset we have is a sampling result from the underlying distribution. The randomness in collecting the dataset for different domains varies a lot and introduces the sampling bias unavoidably (i.e. the sample pattern distributes differently cross domains). Aligning the marginal distributions of the biased datasets will distort the internal structure of the real underlying distribution, leading the underlying conditional distribution of the dataset cannot be same under the bias. Investigating the internal structure of the underlying distribution in DA has been noted by SRDC [49] using discriminative clustering [7], by sample-level matching cross domains [13, 14], or by Optimal Transport Theory[1],etc. But they all neglect the issue mentioned above.

We first elaborate the dilemma mentioned above in Section 2. Then our proposed methodology is presented in Section 3. We introduce the "mirror samples", which are generated to complement the potential bias of the sampled dataset cross domains, making each sample in the domain to have the corresponding instance in the opposite domain. We propose the mirror loss by regularizing the generated mirror sample with the existing sample to achieve more fine-grained distribution alignment. The properties of the proposed methods and the experiment results on several mainstream benchmarks are given in Section 4 and Section 5.

## 2 Delimma of Covariate Shift using Samples

We first revisit the intuition behind the method of reducing the covariate shift. In cross-domain visual classification, the domain distribution can be decomposed into class-discriminative latent visual pattern distribution ($p(x)$) and conditional class distribution ($p(y|x)$). The class-discriminative visual pattern refers to the intrinsic visual characteristics of each class, which is domain-agnostic and follows their natural distribution. The conditional distribution is the probability of those visual patterns belonging to a certain category. Reducing the covariate shift assumes the conditional distributions cross domains are same. Aligning the marginal distribution of visual patterns in certain latent space can achieve the ideal domain alignment.

**Biased Sampling Dataset.** Note that the dataset we have is a sampling result w.r.t. the intrinsic visual property distribution. The sampling process (i.e. dataset collection procedure) varies for different domains and inevitably introduces sampling bias cross domains, resulting the imbalanced patterns within one category etc. One observation of this sampling bias can be seen in the Fig.1(a), where we take the category "Bed" and "Calendar" of domain "Art" and "Clipart" in Office-Home[51] as examples. For the same category, saying "Bed", some of the images are Bunk bed but the others are

not; some of them have pillows while the others do not; even some of them have people on bed while the left are empty etc. However, the ratios of those different visual patterns differ dramatically in the two domains. As shown in Fig.1(a), the ratios of "with People" in Bed are much lower in "Art" than in "Clipart", while the Bunk bed ratio has higher percentage in "Clipart". This difference also exists for "Calendar" in patterns like whether it has digit or not and whether it is daily calendar or monthly calendar. An ideal sampling process should assure the consistent ratios for these patterns following the category's intrinsic distribution, but it's impractical.

**Reducing Covariate Shift vs Assumption Violation.** The biased dataset will induce a dilemma if we still follow the philosophy of reducing covariate shift. Specifically, in the biased dataset, the samples in the source domains do not have the same counterparts in the target domain in distribution. If we align the two domains by the biased data, no matter using moments/prototype alignment or minimizing the sample-based domain discrepancy, we essentially align the samples to biased positions, making the intrinsic conditional probability, which should be same, distorted. Fig.1(b) gives a simplified illustration of this dilemma in one-dimension case. The underlying marginal distributions of $x$ for source and target domains are $U(-3, 5)$ and $U(4, 12)$ respectively, where $U$ means uniform distribution. The conditional distributions of $x$ belonging to certain class in source and target domains are $N_{[-3,5]}(1, 1)$ and $N_{[4,12]}(8, 1)$, where $N_{[a,b]}(\mu, \sigma)$ is truncated normal distribution with support $[a, b]$, $\mu$ and $\sigma$ as mean and standard deviation respectively. This is an ideal case where reducing the covariate shift can perfectly eliminate the domain gap: offsetting the target samples to left by 7 without changing the class probability (from red solid line to green solid line in Fig.1(b)), both the conditional distribution and marginal distribution are same. However, in practice, we only have the random samples illustrated by dots in Fig.1(b). If we still try to reduce the covariate shift, i.e. offsetting the target samples to the source samples in the same order (any order of moments of $x$ are same in this case), the resulting conditional distribution for target domain (the red dashed line in Fig.1(b)) will be distorted, breaking the assumption of covariate shift. This dilemma is what our proposed mirror sample and mirror loss are expecting to reduce.

# 3 Proposed Methods

## 3.1 Preliminaries

**Mirror Samples.** A straightforward solution to reduce the dilemma is to find the ideal counterpart sample in the other domain. Those counterpart sample pairs are in the same positions in their own distribution, i.e. having the equivalent domain-agnostic latent visual pattern. We call it Mirror Sample. If we could find those ideal mirror samples cross domains under the inevitable sampling bias, the alignment by reducing the covariate shift will not incur the conditional inconsistency anymore. The closest work to the mirror sample would be the series of pixel-level GAN based method, such as CyCADA [26] ,CrDoCo[10], etc. However, those methods are dedicated for pixel-level domain adaptation, assuming the domain gap is mainly the "style" difference. But domain shift is generally large and varied. What's more, the adversarial losses might suffer from the mode collapse and convergence issues. The experimental comparison can be found in Appendix E. Different from those ideas, we propose a concise method to construct the Mirror Sample, which consists of Local Neighbor Approximation (LNA) and Equivalence Regularization (ER).

**Optimal transport Explanation**. In fact, the mirror sample can be formulated in terms of optimal transport theory [1]. Let $\mathbb{T}^s$ and $\mathbb{T}^t$ be the two transforms (push-forwards operators) on the two domain distributions $p_{\mathcal{S}}$ and $p_{\mathcal{T}}$ such that the resulting distributions are same, i.e. $\mathbb{T}^s_{\#} p^s = \mathbb{T}^t_{\#} p^t$. This sheds light on an elegant way to reduce the domain gap. In this context, $x^s \in D^{\mathcal{S}}$ and $x^t \in D^{\mathcal{T}}$ are the mirror for each other if $\mathbb{T}^s_{\#} p^s(x^s) = \mathbb{T}^t_{\#} p^t(x^t)$. Direct finding the push-forwards operators are almost impossible. However, if we would first find the mirror samples defined above, those operators would like be learned by those mirror constraints. The more detailed explanation and illustration can be found in Appendix A.

**Denotations.** Similar to the existing settings of DA, denote the source samples as $\{(x_i^s, y_i^s)\}_{i=1}^{n^s}$, $y_i^s \in \mathcal{Y}$ and the target samples as $\{x_j^t\}_{j=1}^{n^t}$, where $n^s$ and $n^t$ are the numbers of source and target samples, $\mathcal{Y}$ is the label set with $M$ classes. We also use $X^{\mathcal{S}} = \{x_i^s\}_{i=1}^{n^s}$, $X^{\mathcal{T}} = \{x_j^t\}_{j=1}^{n^t}$ for brevity. The notations with "tilde" above refer to the mirror samples. Since the proposed method follows the

framework of domain-invariant representation, all the above notations as well as the mirror samples are in the latent space after certain transformation.

## 3.2 Local Neighbor Approximation

The Local Neighbor Approximation (LNA) is expected to generate mirror samples using the local existing data in the same domain. Considering that if the source and target domains are aligned ideally, the source and target domains are two different views of the same distribution. The mirror pairs, although in different domains, are exactly the same instance in the aligned space. Inspired by $d$-SNE in [57], we use the nearest neighbors in the opposite domain to estimate the mirror of a sample. Fig.2 gives an illustration of the local neighbor approximation cross domains. Formally, denote $d$ as the distance measure of two samples. To construct the mirror of target sample $x_j^t$, we first find the nearest neighbor set in the source domain as $\tilde{X}^{\mathcal{S}}(x_j^t)$, called mirror sets as follows:

$$\tilde{X}^{\mathcal{S}}(x_j^t) = \arg \top_{x \in X^{\mathcal{S}}}^k d(x, x_j^t) \tag{1}$$

where $\top_{\Omega}^k$ is a "top-$k$" operation that selects the top $k$-smallest elements in set $\Omega$ with respect to the distance measure $d$. Then we estimate the mirror sample of $x_j^t$ by weighted combination of the samples in $\tilde{X}^{\mathcal{S}}(x_j^t)$. This is following the conclusion that the learned features after the feature extractor lie in a manifold [3, 57]:

$$\tilde{x}^s(x_j^t) = \sum\nolimits_{x \in \tilde{X}^{\mathcal{S}}(x_j^t)} \omega(x, x_j^t) x \tag{2}$$

where $\omega(x, x_j^t)$ is the weight of the element $x$ in the mirror set $\tilde{X}^{\mathcal{S}}(x_j^t)$. The weight can be inversely proportional to t $d(x, x_j^t)$ or simply $1/k$. Symmetrically, we can also have the corresponding mirror of source sample $x_i^s$ in target domain as $\tilde{x}^t(x_i^s)$ analogous to Eq.1 and 2.

## 3.3 Equivalence Regularization

Although LNA provides a way to estimate the mirror sample, it cannot guarantee the equivalence of the mirror pairs. We propose an anchor-based Equivalence Regularization (ER) method to enhance the equivalence cross domains.

In detail, define the centers for class $c$ in source and target domains as $\mu_c^s$ and $\mu_c^t$ respectively. If the distributions are aligned, those class-wise centers for the same class should be same. This means they could be the anchors cross domains. Inspired by the probability vector used in SRDC [49], we introduce the relative position of sample $x_j^t$ to an anchor $\mu_c^t$ of its domain as:

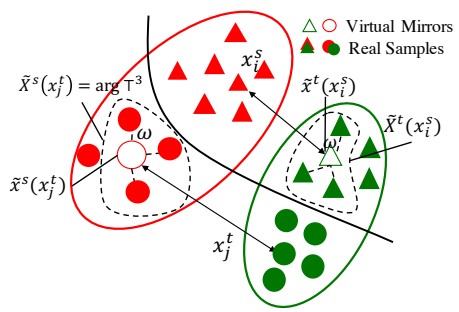

Figure 2: Local Neighbor Approximation: the mirror sets $\tilde{X}^s$ and $\tilde{X}^t$ are calculated by Eq.(1) using virtual mirrors. We show the case when we set $k = 3$.

$$q_c^t(x_j^t) = \frac{\exp\{-d(x_j^t, \mu_c^t)\}}{\sum_{c=1}^M \exp\{-d(x_j^t, \mu_c^t)\}} \tag{3}$$

where $d$ is a distance measure. Then the relative position vector w.r.t. all the anchors is

$$q^t(x_j^t) = \left[ q_1^t(x_j^t), q_2^t(x_j^t), \cdots, q_M^t(x_j^t) \right] \tag{4}$$

where $M$ is the class number. For $x_j^t$'s mirror sample $\tilde{x}^s(x_j^t)$ estimated by LNA, its relative position to its anchors $\mu_c^s$ is symmetrically written as:

$$q_c^s(\tilde{x}^s(x_j^t)) = \frac{\exp\{-d(\tilde{x}^s(x_j^t), \mu_c^s)\}}{\sum_{c=1}^M \exp\{-d(\tilde{x}^s(x_j^t), \mu_c^s)\}} \tag{5}$$

Its relative position vector w.r.t. to all source anchors is analogously written as:

$$q^s(\tilde{x}^s(x_j^t)) = \left[ q_1^s(\tilde{x}^s(x_j^t)), q_2^s(\tilde{x}^s(x_j^t)), \cdots, q_M^s(\tilde{x}^s(x_j^t)) \right] \tag{6}$$

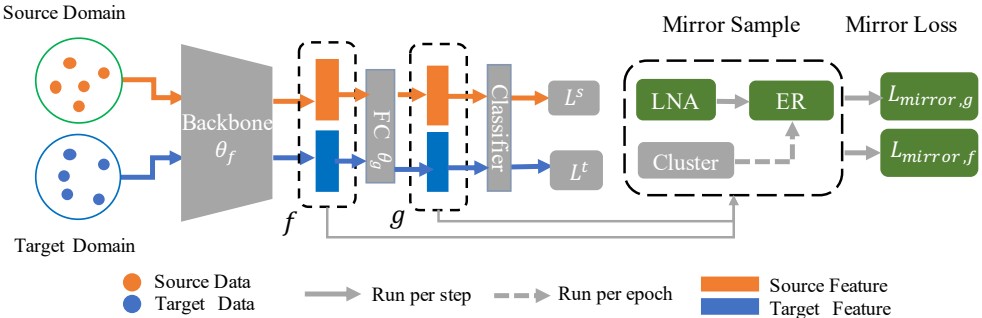

Figure 3: Overall structure of the model: the LNA and ER are applied to the output of the backbone and the first FC layer to calculate the mirror and mirror loss. The detailed algorithm can be found in Appendix.

Note that Eq.3, 4, 5 and 6 can be used to get the corresponding $q_c^s(x_i^s)$ and $q^s(x_i^s)$, $q_c^t(\tilde{x}^t(x_i^s))$ and $q^t(\tilde{x}^t(x_i^s))$ by switching the source and target domains.

To regularize the equivalence of the mirror pairs, we minimize the Kullback-Leibler divergence of the relative position vectors of them, forming the mirror loss:

$$\mathcal{L}_{mr,x} = \frac{1}{n^t} \sum_{j=1}^{n^t} \text{KL}\big(q^t(\tilde{x}^s(x_j^t)) \| q^t(x_j^t)\big) + \frac{1}{n^s} \sum_{i=1}^{n^s} \text{KL}\big(q^s(\tilde{x}^t(x_i^s)) \| q^s(x_i^s)\big) \qquad (7)$$

where the first term of right side of the equation is aligning the relative position vector of target sample $x_j^t$ and its virtual mirror $\tilde{x}^s(x_j^t)$ in source domain, while the second term is aligning the source sample $x_i^s$ to its virtual mirror $\tilde{x}^t(x_i^s)$ in target domain.

One should note that although some existing works use KL divergence losses such as TPN [37] or SRDC [49] etc., mirror loss is quite different. The mirror loss minimizes the divergence of the relative position vectors of *mirror pairs* to the anchors, ensuring the constructed sample by LNA to be the equivalent. The other methods do not involve constructed samples. They generally minimized the KL divergence of different distributions for the *same sample*.

### 3.4 Model with Mirror Loss

The mirror samples actually augment the existing datasets to $\{x_i^s\}_{i=1}^{n^s} \cup \{\tilde{x}^s(x_j^t)\}_{j=1}^{n^t}$ for source and $\{x_j^t\}_{j=1}^{n^t} \cup \{\tilde{x}^t(x_i^s)\}_{i=1}^{n^s}$ for target. The mirror loss setups the constraints cross domains between $\{\tilde{x}^s(x_j^t)\}_{j=1}^{n^t}$ and $\{x_j^t\}_{j=1}^{n^t}$, $\{x_i^s\}_{i=1}^{n^s}$ and $\{\tilde{x}^t(x_i^s)\}_{i=1}^{n^s}$, rather than the existing dataset, expecting to relieve the sampling bias and the mentioned dilemma.

Fig.3 illustrates the model structure we use here. After the backbone we have the feature $f \in \mathbb{R}^{d_f}$, then we use additional one full-connected layer to have feature representation $g \in \mathbb{R}^{d_g}$, which is finally fed into the final classifier. We incorporate the mirror loss by applying LNA and ER to the source and target features $f$ and $g$, resulting the mirror loss $\mathcal{L}_{mr,f}$ and $\mathcal{L}_{mr,g}$. For the labeled source data $\{(x_i^s, y_i^s)\}_{i=1}^{n^s}$, we use cross entropy loss, i.e. $\mathcal{L}^s = -\frac{1}{n^s} \sum_{i=1}^{n^s} \sum_{c=1}^{M} \mathbb{I}(y_i^s = c) \log p_{i,c}^s$, where $p_{i,c}^s$ is the predicted probability for class $c$ and $\mathbb{I}$ is the indicator function. For the unlabeled target data $\{x_j^t\}_{j=1}^{n^t}$, we follow the unsupervised discriminative clustering method [27] to introduce an auxiliary distribution as soft pseudo label. Then we have the target-related loss as $\mathcal{L}^t = -\frac{1}{n^t} \sum_{i=1}^{n^t} z_i^t \cdot \log p_i^t$, where $p_i^t \in \mathbb{R}^M$ is the predicted probability using current learned network and $z_i^t \in \mathbb{R}^M$ is the auxiliary distribution with each entity as $z_{i,c}^t \propto \frac{p_{i,c}^t}{(\sum_{i=1}^{n^t} p_{i,c}^t)^{1/2}}$. Note that $z_i^t$ will be updated once for each epoch rather than iterating until convergence in [27]. To sum up, the total loss of the model is:

$$\mathcal{L} = \mathcal{L}^s + \mathcal{L}^t + \gamma(\mathcal{L}_{mr,f} + \mathcal{L}_{mr,g}) \qquad (8)$$

## 4 Asymptotic Properties of Mirrors

We present the theoretical analysis for the mirror sample-based alignment in terms of target error $\mathcal{R}_\mathcal{T}(h)$ and source error $\mathcal{R}_\mathcal{S}(h)$ following the theoretical framework of [2]. If the mirror-based

alignment empirically aligns the underlying distribution cross domains (Proposition 1), we could have lower target error asymptotically (Proposition 2). The proofs are given in Appendix C.

**Proposition 1.** *Denote* $\Phi_{\mathcal{S}}(x), \Phi_{\mathcal{T}}(x)$ *as the density function for domain* $\mathcal{S}$ *and* $\mathcal{T}$*, with supports as* $D^{\mathcal{T}}$ *and* $D^{\mathcal{S}}$ *respectively.* $\mathcal{H}$ *as the hypothesis class from features to label space. If* $\Phi_{\mathcal{S}}(x) \overset{a.s.}{=} \Phi_{\mathcal{T}}(x)$*, then* $d_{\mathcal{H}\Delta\mathcal{H}}(\mathcal{S}, \mathcal{T}) \to 0$*, where* $d_{\mathcal{H}\Delta\mathcal{H}}(\mathcal{S}, \mathcal{T}) = 2\sup_{h,h'\in\mathcal{H}} \big| \Pr_{x\sim D^{\mathcal{S}}}[h(x)\neq h'(x)] - \Pr_{x\sim D^{\mathcal{T}}}[h(x)\neq h'(x)] \big|$*.*

Proposition 1 states that the distribution alignment for certain learnable space will reduce the domain discrepancy in terms of functional differences $d_{\mathcal{H}\Delta\mathcal{H}}$. The above distribution alignment can be achieved *empirically* by minimizing the $\mathcal{L}_{mr,x}$ in Eq.(7). From the definition, we can see that $\mathcal{L}_{mr,x}$ is minimized if and only if $q^s(\tilde{x}^s(x_j^t)) = q^t(x_j^t)$ and $q^t(\tilde{x}^t(x_i^s)) = q^t(x_i^s)$ for every mirror pairs for $x_i^s$ and $x_j^t$. It means: 1) the class centers (anchors) for both source and target domain are same, 2) the mirror pairs cross domains have the same position relative to the common centers $\mu_c, c = 1, 2, \cdots, M$. Thus the empirical density function $\hat{\Phi}_{\mathcal{S}}(x)$ and $\hat{\Phi}_{\mathcal{T}}(x)$ over $X^{\mathcal{S}}$ and $X^{\mathcal{T}}$ are same, i.e. $\hat{\Phi}_{\mathcal{S}}(x) = \hat{\Phi}_{\mathcal{T}}(x)$. Since $X^{\mathcal{S}}$ and $X^{\mathcal{T}}$ are sampled from $D^{\mathcal{S}}$ and $D^{\mathcal{T}}$ w.r.t. the underlying density function $\Phi_S$ and $\Phi_T$, Glivenko–Cantelli theorem [41] could assure that when $n^t, n^s \to \infty$, we have

$$\Phi_{\mathcal{S}}(x) \overset{a.s.}{=} \hat{\Phi}_{\mathcal{S}}(x) = \hat{\Phi}_{\mathcal{T}}(x) \overset{a.s.}{=} \Phi_{\mathcal{T}}(x) \tag{9}$$

where $a.s.$ means almost surely. So based on Proposition 1, minimizing $\mathcal{L}_{mr,x}$ will reduce $d_{\mathcal{H}\Delta\mathcal{H}}(\mathcal{S}, \mathcal{T})$ to zero empirically when the number of samples is large.

**Proposition 2.** *Define* $\lambda = \min_{h\in\mathcal{H}}\{\mathcal{R}_{\mathcal{S}}(h, h_{\mathcal{S}}) + \mathcal{R}_{\mathcal{T}}(h, h_{\mathcal{T}})\}$ *same to [2], where* $h_{\mathcal{S}}$ *and* $h_{\mathcal{T}}$ *are the labeling functions in each domain. Denote* $\lambda_m + \frac{1}{2}d_{\mathcal{H}\Delta\mathcal{H}}^m$ *as the term of* $\lambda + \frac{1}{2}d_{\mathcal{H}\Delta\mathcal{H}}$ *when* $\mathcal{L}_{mr,x}$ *is minimized. If minimizing* $\mathcal{L}_{mr,x}$ *aligns the distribution in the learned space, we have*

$$\lambda_m + \frac{1}{2}d_{\mathcal{H}\Delta\mathcal{H}}^m \leq \lambda + \frac{1}{2}d_{\mathcal{H}\Delta\mathcal{H}} \tag{10}$$

Note that $\lambda + \frac{1}{2}d_{\mathcal{H}\Delta\mathcal{H}}$ is the main gap between source and target error stated in [2]. Proposition 2 indicates that when the mirror loss is minimized, we would get a lower gap. The key insight behind proposition 2 is that if the discrepancy of the underlying distribution is empirically approaching to 0, we can have a more relaxed effective hypothesis of $\mathcal{H}$, leading to a lower value of $\lambda$. This advantage can be obtained by mirror loss.

## 5 Experiments

### 5.1 Datasets and Implementations

**Datasets**. We use **Office-31** [42], **Office-Home**[51], **ImageCLEF** and **VisDA2017**[39] to validate our proposed method. Office-31 has three domains: Amazon(A), Webcam(W) and Dslr(D) with 4,110 images belonging to 31 classes. Office-Home contains 15,500 images of 65 classes with four domains: Art(Ar), Clipart(Cl), Product(Pr) and RealWorld(Rw). ImageCLEF contains 600 images of 12 classes in three domains: Caltech-256(C), ILSVRC 2012(I) and Pascal VOC 2012(P). VisDA2017 contains $\sim$280K images belonging to 12 classes. We use "train" as source domain and "validation" as target domain.

**Implementations**. We implement our model in PyTorch. We use ResNet50 [24] or ResNet101 pre-trained on the ImageNet as backbone shown in Fig.3. The learning rate is adjusted by $\eta_p = \eta_0(1 + \alpha p)^{-\beta}$ like [17], where $p$ is the epoch which is normalized in [0, 1], $\eta_0 = 0.001$, $\alpha = 10$ and $\beta = 0.75$. The learning rate of fully connected layers is 10 times of the backbone layers. When calculating the centers $\mu_{f,c}^t$ and $\mu_{g,c}^t$, we use the class-wise centers of source domain $\mu_{f,c}^s$ and $\mu_{g,c}^s$ as the initial centers in $k$-Means clustering. To enhance the alignment effect, we use the centers $\mu_{f,c} = 0.5\mu_{f,c}^s + 0.5\mu_{f,c}^t$ and $\mu_{g,c} = 0.5\mu_{g,c}^s + 0.5\mu_{g,c}^t$ in calculating the mirror loss in Eq.(7). To estimate the virtual mirror using LNA, the additional operation "Top-$k$" can be implemented using priority queue of length $k$. It only brings additional $O(n^s)$ and $O(n^t)$ computation costs during training. The virtual mirror weight, i.e. $\omega(w, x_j^t)$ is $1/k$. All the experiments are carried out on one Tesla V100 GPU.

Table 1: Comparsion with SOTA methods(%). All the results are based on ResNet50 except those with mark $^\dagger$, which are based on ResNet101. Red indicates the best result while Blue means the second best. The mark $^*$ means the result is reproduced by the offically released code.

| Method | Office-31 | Office-Home | CLEF | VisDA2017 |
|---|---|---|---|---|
| Source Model[24] | 76.1 | 46.1 | 80.7 | 52.4$^\dagger$ |
| MDD[60] | 88.9 | 68.1 | 88.5$^*$ | 74.6 |
| JDDA[7] | 80.2 | 58.5$^*$ | 83.3$^*$ | 62.5$^{*\dagger}$ |
| MCSD[59] | 90.7 | 69.6 | 90.0 | 71.3 |
| SAFN [56] | 87.1 | 67.3 | 88.9 | 76.1$^\dagger$ |
| CAN [28] | 90.6 | 68.8$^*$ | 89.6$^*$ | 87.2$^\dagger$ |
| RSDA-MSTN[22] | 91.1 | 70.9 | 90.5 | 75.8 |
| SHOT[32] | 88.7 | 71.6 | 87.2$^*$ | 79.6$^{*\dagger}$ |
| SRDC[49] | 90.8 | 71.3 | 90.9 | — |
| BSP-TSA[9] | 90.6 | 71.2 | 88.9$^*$ | 82.0 |
| FixBi[36] | 91.4 | 72.7 | 86.0$^*$ | 87.2$^\dagger$ |
| Ours | **91.7** | **73.4** | **91.6** | **87.9**$^\dagger$ |

Table 2: Ablation studies using Office-Home and Office-31 datasets based on ResNet50.($K = 3$)

| Baseline | FC Mirror | Bk Mirror | Office-Home | Office-31 |
|---|---|---|---|---|
| ✓ | | | 66.5 | 85.5 |
| ✓ | ✓ | | 71.7 | 89.7 |
| ✓ | | ✓ | 71.8 | 90.0 |
| ✓ | ✓ | ✓ | **73.4** | **91.7** |

Table 3: Parameter Sensitivity Analysis for $k$

| Parameters | | Office-Home | Office-31 |
|---|---|---|---|
| | 1 | 71.7 | 90.2 |
| | 3 | **73.4** | **91.7** |
| $k$ | 5 | 71.8 | 90.3 |
| | 7 | 71.3 | 89.8 |
| | 9 | 71.4 | 89.8 |

## 5.2 Comparison with SOTA Results

We compare our methods with several types of SOTA methods under the same settings on the four datasets. They include the methods using maximum mean discrepancy and its variants: e.g. DAN [34], JDDA [7], SAFN [56], MDD [56] etc; class-wise center/centroid based methods: e.g. CAN [28]; adversarial learning related methods, e.g. DANN [17], ADDA [50] etc; as well as the most recent methods such as MCSD [59], SRDC [49] etc.

Table 1 shows the average results of all the tasks. Detailed results for each task can be found in supplementary material. We can find that our method has made a consistent and significant improvement over the existing SOTA methods. For the relatively simple datasets Office-31, our method improves by 0.3% comparing with SOTA. For the more challenging dataset Office-Home and the large-scale dataset VisDA2017, our method improves about 0.7% on average.

## 5.3 Model Analysis

We take Office-Home and Office-31 as examples to investigate the different components of the proposed model. Average results for all tasks are presented. Detailed results are given in the appendix.

**Ablation Study**. To investigate the efficacy of the proposed mirror loss, we experiment several model variations. The "Baseline" model only uses the backbone and the source and target losses, i.e. $\mathcal{L}^s + \mathcal{L}^t$ in Eq.(8). Then the mirror loss is added gradually. Specifically, we applied the mirror loss to the last pooling layer of the backbone, i.e. feature layer $f$ in Session 3.4 ("BK Mirror"), the output of first full-connected layer after the backbone, i.e. the feature layer $g$ in Session 3.4 ("FC Mirror") and finally both. Table 2 gives the results when $k = 3$ and $\lambda = 1.0$. From the results, we can see that by adding the mirror loss, the performance will boost at least 5.2% on Office-31 and 4.2% on Office-Home. The difference between the "FC Mirror" and "BK Mirror" is small. When we apply the mirror loss to both layers $f$ and $g$, the final results can achieve SOTA on the datasets. In fact, only using losses on the source and target domains neglects the connection between the underlying distributions cross domains, which is what the mirror loss focuses on.

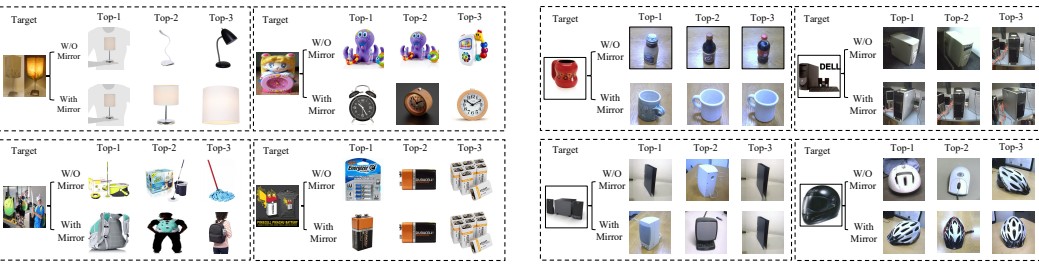

Figure 4: Visualization of mirror sets for Office Home: the source domain is "Product" and the target domain is "Art".

Figure 5: Visualization of mirror sets for Office 31: the source domain is "Webcam" and the target domain is "Amazon".

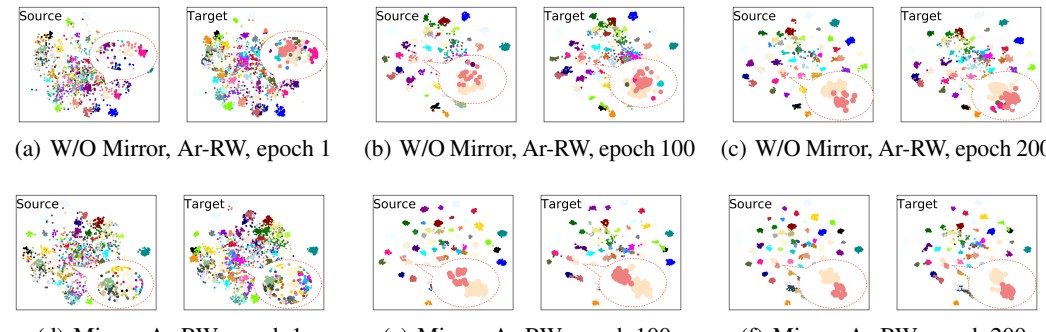

(a) W/O Mirror, Ar-RW, epoch 1    (b) W/O Mirror, Ar-RW, epoch 100    (c) W/O Mirror, Ar-RW, epoch 200

(d) Mirror, Ar-RW, epoch 1    (e) Mirror, Ar-RW, epoch 100    (f) Mirror, Ar-RW, epoch 200

Figure 6: Visualization of cluster evolvement for task Ar-Rw in Office-Home. Different classes are distinguished by color.

**Sensitivity Analysis**. The parameter $k$ in Eq.(1) controls how we construct the virtual mirrors. A larger $k$ means choosing a larger mirror set and it may lead to more indistinguishable mirrors. A smaller $k$ may get an unreliable or even wrong mirror. In Table 3, we investigate the accuracies with different $k$s by setting $k = 1, 3, 5, 7, 9$. In terms of average accuracy, we can see that $k = 3$ is the best choice. The parameter $\gamma$ in Eq.(8) controls the weight of the mirror loss. We tried different values from 0.0 to 3.0 to investigate the best choice for different tasks. Note the $\gamma = 0.0$ means we do not use mirror loss. Table 3 also gives the average results on both Office-31 and Office-home datasets. We can see that different tasks have different optimal $\gamma$s (refer to the supplementary material). In most of the tasks, the optimal $\gamma$ is in range $1.0 \sim 2.0$ for the two datasets. Besides above, we also evaluate the sensitivity of the key components to construct the mirror samples in Eq.2, i.e. the weight of $w$ and the selection of $d$. We use Euclidean and Gaussian-kernel distance, combined with the weight of $1/k$ and inverse proportion of the distance. The final results varied not much on Office-Home, from 72.4 to 73.4, indicating that these components are robust to the variations. Detailed results are in Appendix E.

**Visualization of Mirror Set** To further illustrate the virtual mirror in real datasets, we visualize the mirror set defined in Eq.(1) in Fig.4 and 5. We use the embeddings in layer $f$ to find the top-3 similar samples in the source w.r.t. to certain sample in target domain. As a comparison, the same top-3 similar samples by the model without mirror loss are also given. Overall, we can see that the top-3 similar samples with mirror loss are more similar compared to the results without mirror loss. Specially, the "clock" class in Fig.4 might consist of quite dissimilar samples although they belong to the same class for the model without mirror loss. In Fig.5, the "Bottle" has mirror sets belonging to the same class, but our proposed method gives results much more similar. For the "Helmet", the results without mirror loss consist of even different class samples, such as "mouse".

**Impacts of the Biased Dataset**. As pointed out in Section 2, the visual patterns are biased among the source and target domains, which will incur the dilemma of domain alignment. To elaborate how our proposed methods solve this issue, we compare the error rates of each visual pattern using the mirror samples or not. Table 4 gives the results for "Bed", "Calendar" and "Bucket" in the task of "Ar $\rightarrow$ Cl" in Office-Home. For the category "Bed", there is no "Bunk bed" for domain Ar while 7.1% for Clipart. Among the error-predicted samples, the "Bunk bed" consists of 2.0% for our proposed

Table 4: Error Rates for visual patterns of task Ar → Cl in Office-Home. "Source/Target" refers to the ratios (%) of the visual pattern in source and target domains. "W/O Mirror" gives the error rates (%) using the baseline model without mirror loss.

| Category | Bed | | | Calendar | | Bucket | |
|---|---|---|---|---|---|---|---|
| Visual Pattern | Bunk Bed | With People | With Pillow | Daily Calendar | W/O. Digit | With Brush | W/O. People |
| Source/Target | 0.0/7.1 | 2.5/20.4 | 12.5/14.3 | 0.0/20.5 | 15.0/42.6 | 0.0/37.0 | 82.5/100 |
| W/O Mirror | 8.5 | 28.8 | 59.0 | 42.0 | 31.0 | 42.0 | 57.0 |
| Mirror | 2.0 | 14.9 | 51.0 | 28.5 | 24.0 | 25.8 | 38.0 |

method, which is much lower than 8.5%, the results of the baseline method without using mirror loss. Similar observation can be found for other patterns. These results directly validate that the proposed methods alleviate the bias of the performance resulting from the biased dataset.

**Visualization of embeddings**. We visualize the alignment procedure by t-SNE [25] of the feature embeddings. We record both source and target features in layer $f$ during different training epochs. Then t-SNE is carried out for all the features at different epochs once to assure the same transformation is applied to them. Fig.6 shows the detailed evolvement of the different classes and distributions for the task of Ar to Rw in Office-Home. As a comparison, the evolvement for the model without mirror loss is also given. At beginning, the samples of source and target are messy. As the training progressing, such as epoch 100 in Fig.6(b) and Fig.6(e), the cluster discriminality and the cluster shape are clearer and more consistent. At the final stage, i.e. epoch 200, our proposed method has much more "shape" similarity between source and target domains. This reflects the proposed mirror and mirror loss have achieved higher consistency between the underlying distributions.

# 6   Related Works

Researchers have designed substantial approaches to eliminate the domain gap, from model side or data side. From the model side, existing methods try to align the two domains using different terms of discrepancy metrics, like maximum mean discrepancy (MMD) [21, 34], differences of the first- or high-order statistics of distributions [47, 58, 8], inter or intra class distance [7, 57], and even Geodesic Flow Kernel [52, 18] in Grassman manifold. Furthermore, to enhance alignment effects, adversarial learning between domain discriminator and classifier is also widely investigated, like gradient reversal layer in DANN [17], ADDA [50], conditional adversarial learning CDNN[35], 3CATN [30], multiple classifiers discrepancy [43] and batch spectral penalization [9] etc. There are also the meta-learning[53], disentangling learning [29] and dynamic weighting [54] following this line. The manipulation of data in cross domain alignment provides a strong complement to the model design. They include techniques like generating virtual or intermediate domain that boosts the performance by reducing the difficulty the model is facing [55, 12, 36, 37], data augmentation [31] that mimics the target domain. Inspired by GAN [19], generating the sample directly is also an effective way for domain adaptation to enhance the model capability cross domains. Typical works are PixelDA [5], CoGAN[33], CyCADA[26] and CrDoCo[10] etc. They generally work on the pixel-level domain gap like style or depth differences. In sum, almost all the methods are working to reduce covariate shift implicitly or explicitly.

As far as we know, there is few researches discussing the sample-level domain alignment, not alone considering the biased dataset. One type of the most related works is the methods using sample-level domain alignment. SRDC [49] used samples to form structure regularization for each domain. [16, 11, 13] took advantage of a simplified discrete version of optimal transport theory. [37] regularized the distance distribution of each sample for different prototypes. However, they do not consider the biased issue discussed in Section 2. The other type is the generative method like CyCADA [26] etc. Although generating virtual sample in target domain is like mirror sample in our methods, those work only apply to the pixel-level task or the classification tasks with strict domain gap assumption. Those methods heavily rely on GAN-based method, which is complicated comparing to our method and suffers from mode collapse and divergence. In fact, aligning the distribution by sample data can trace back to the work of [46, 20], but there is no further discussion in the UDA field.

# 7 Conclusion

In this paper, we uncover the dilemma when using the sampled data to reduce the covariate shift in unsupervised domain adaptation and further propose the mirror sample and mirror loss to solve this issue. The mirror sample is constructed using the local neighbor approximation and equivalence regularization, expecting to approximate equivalent samples in the marginal distributions. The ablation analysis as well as the visualizations demonstrate the efficacy of the proposed mirror and the mirror loss. Current construction method of mirror samples is a start point, which would have brittle improvements when the dataset is extremely biased or sparse. It is also limited to the visual classification task. More sophisticated and generic methods are expected. What's more, we believe that both the dilemma and the idea of using equivalent sample for distribution alignment should have more discussion in both domain adaptation and transfer learning.

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
