# OpenReview forum: "Reducing the Covariate Shift by Mirror Samples in Cross Domain Alignment"
_NeurIPS.cc/2021/Conference — NeurIPS 2021 Poster_

### Official Review · Reviewer_ZzRU · 2021-07-03

**Rating:** 6
**Confidence:** 4

**Summary:**

In this paper, the author proposed a sample-level alignment method for solving the covariate shift problem in unsupervised domain adaptation. They noticed that the existing methods do not work well since they neglect the bias in the datasets (e.g. different visual patterns in the same category). To fix this problem, for each sample in the source/target domain, they construct a mirror sample in the other domain by a weighted average of the top K nearest neighbors. They also derived new losses to preserve the geometric structures of latent representations in the subspace. The overall idea is straightforward but interesting. The experimental results achieve SOTA performance on public benchmarks.

**Limitations And Societal Impact:**

The authors have addressed the limitations and potential negative societal impact of their work.

**Main Review:**

technical concerns:
- In fig.1(b) one-dimension case, this example is somehow misleading. It is trying to convey the message that distribution alignment is equivalent to sample alignment. One may fail to align two distributions due to the different number of samples drawn from each distribution. This example is not convincing.
- In fig.2, by the definition of mirror set in line 121 eq(1) (let's consider Euclidean distance), for each sample in source, its corresponding mirror set will always consist of samples close to the left boundary, for instance, blue triangles in the dashed circle. Is this true for the 2D example in fig.2?
- According to line 223-225 and Algorithm 1 in the appendix, the centers of the target are computed based on the pseudo labels that generated from the classifier. Since the classifier may produce wrong labels for target examples, these target centers can be inaccurate. The mirror losses are highly dependent on these inaccurate centers to promote the latent representations preserve geometric structures in the subspace. Does the mirror sample still a semantic counterpart? Will their labels change from one category to another? Will this introduces extra noise for supervised learning (classification task)? Is there any strategy to handle these problems?

*Originality*: The novelty is limited. As the author claimed, the main contribution of this paper is the idea of mirror samples and mirror losses. CyCADA uses GAN to generate a 'mirror sample' in the target domain for each sample from the source domain, while in this paper mirror samples were constructed by inquiring neighbors. I am quite interested in the comparison between these two approaches. However there is no comparison experiments for these two methods.

 *Quality*: The proposed method is well motivated. Their claims have theoretical analysis and are supported with good experimental results. But most ideas rely on heuristic knowledge. Considering the aforementioned technical concerns, the proposed method for constructing mirror samples is not principled.

*Clarity*: The paper is easy to follow. But there are typos and grammar errors, e.g., line 146 eq(3),  $\mu_c^s$ should be $\mu_c^t$ .

*Significance*: The significance of this submission is moderate.

**Time Spent Reviewing:**

20

---

> ### Author Response · Authors · 2021-08-10
> **Responses to Reviewer ZzRU**
>
> We appreciate the reviewer to give detailed and insightful comments about our work.
> We hope our following responses can relieve some misunderstanding or present our complementary analysis about this work, making the reviewer to re-evaluate the work with better impression.
>
> #### 1. Concerns about Fig1(b).
> We're sorry to bring confusion. In fact, the failure of distribution alignment originates from the randomness of the sampling, not the number of samples.
> In Fig1(b), the number of samples are same.
> We'll improve the statements to avoid the confusion in the final version.
>
> #### 2. Concerns in Fig.2
> The dashed circle means the mirror set expressed in eq(1).
> Fig.2 is used to illustrate how to get mirror sets and how to generate mirror samples using mirror sets in 2d case.
>
>
> #### 3. Discussion about the pseudo-label:
> This is an insightful question.
> The pseudo-label plays an important role in our methods.
> To enhance the reliability of the target clusters, the following two strategies are used.
> 1) We use the k-means cluster, instead of classifier to assign the pseudo labels.
> Using the k-means cluster with the source class centers as the initialized centers (line 223-228).
> Since the K-Means cluster has stability guarantees(an Expectation-Maximation scheme), it can provide the local optimal assignment for the class considering reliable source data centers(they're based on labels).
> Also, only use the centers is asymptotically has less noise than sample level pseudo-labels.
> 2) In implementation, we further combine both source center and target center as stated in line 225-226 as the real centers in ER, which can alleviate the unreliable of them.
> We have to admitted that the reliable pseudo-label generation still lacks theory understanding in community. It's effects in transfer learning, semi-supervised learning are future work.
>
>
>
> #### 4. Discussion about CyCADA
> CyCADA is an excellent work for domain adaptation, which generates virtual target samples by transferring the style from source to target domain.
> We could compare our proposed method with CyCADA from two aspects
>
> 1) In concept as CyCADA stated, CyCADA is similar to mirror sample, but is dedicatedly designed for image ''style'' transfer, which is only one specific form of domain shift.
> It would have more limitations when generating sample from one domain to another when the domain gap is large and varied, like generating the distorted sketch from real world photos.
> Our proposed method, from another viewpoint, uses the samples of the target domain to generate the mirror sample.
> By definition, the virtual sample generation does not presume the form of the domain shift.
> The equivalence was assured by ER scheme.
>
>
> 2) In implementations, our network is more concise and easier to train, while CyCADA used several adversarial losses. Its training involves alternative training between generator and discriminator, which needs more hyper-parameter tuning and might suffer more from mode collapse or convergence issues.
> Its performance also depends on the quality of any of the key parts, including determinator(D), style transform modules ($G_{s\to t}$ and $G_{t \to s}$).
>
>
> 3) Experimentally, CyCADA as well as its following work mainly show the results in pixel-level tasks. There are no released code and the results of CyCADA in traditional UDA benchmarks such as Office-31, Office-home.
> That's why we did not provide the experiment comparison in the original submission.
> Even though, we compare our proposed mirror loss with CyCADA on Digits. The results are followed.
> We can see the mirror outperforms CyCADA in Digits by large margin.
>
> | Methods |   USPS->MNIST | SVHN->MNIST |   MNIST->USPS|
>
> Mirror  | 98.5          | 92.1             | 98.7|
>
> CyCADA |  96.5    |       90.4  |            95.6|
>
> We have tried to reimplement CyCADA in Office-31. Unfortunately until recent days, it failed with average accuracy around 68%,a large gap comparing with SOTA methods.
>
>
> #### 5. Concerns about ''heuristic knowledge''
> We start the paper by giving illustrative examples in Fig1(a) and (b), hoping the readers be easier acknowledged by our motivations(in order to keep the main paper from too much theory details)
>
> However, the idea is actually motivated by the optimal transport theory[1] and follows the line of distribution alignment methods [2](Part A of the appendix).
> In details, aligning the two distributions involves designing the pushforward operator to make both distributions equal almost everywhere. i.e. the same position should have the same probability.
> If this can be achieved, the target risk can be reduced as stated in Proposition 1 of the main paper with proofs in appendix.
>
> To this end, we resort to the learnable backbone network as the push-forward operator.
> The left key issue would be how to define the objective considering the "same position covariate should have the same probability".
> Then we derive LNA and ER.
> The proposed mirror construction method is designed to find the corresponding covariate of the distribution to make the alignment.
> The asymptotic properties using LNA is presented in Proposition 2 with proofs in appendix, which states that using LNA and ER would reduce the target risk.
>
> In sum, the ideas in fact are inspired by the above theory and related work. We expect these can relieve the reviewer's concerns.
>
>
> [1] Computational optimal transport. Foundations and Trends in Machine Learning, 11(5-6):355–607, 2019.
>
> [2] Shai Ben-David,et al. A theory of learning from different domains. Machine learning, 79(1-2):151–175, 2010.

---

> > ### Comment · Reviewer_ZzRU · 2021-08-18
> > **Thanks for the clarifications**
> >
> > I'd like to thank the authors for addressing my concerns. I like your work, which is well-motivated and interesting. But it also takes me a long time to think about how does the proposed method work under some extreme cases. I don't mean to raise the bar here, actually I hope we can get better insight into your idea and enjoy the discussion.
> >
> > - When you define the $q(.)$ function in Eq4/6, there is a superscript to indicate the anchors are either from source or target. However, in Eq7, the superscripts are discarded which can lead to confusions. To my understanding, the first term in Eq7 should be $KL(q^s(\tilde{x}^s(x_j^t))||q^t(x_j^t))$. Is this correct? or it is supposed to be $KL(q^s(\tilde{x}^s(x_j^t))||q^s(x_j^t))$.
> >
> >
> > - I think one of your major motivations is to make each sample in one domain have the corresponding instance in terms of similar semantic meaning or visual patterns in the other domain. Since it is almost impossible to find the counterpart sample from the other domain, LNA and ER are used for approximating the mirror samples. Let's consider a toy case, for the "calendar" category,  the source domain contains daily calendar only and the target domain consists of monthly calendar only. Given a monthly calendar from the target, LNA will find a mirror set in the source which consists of daily calendar samples. And it will approximate the mirror sample of monthly calendar as a weighted sum of daily calendar samples. Is the approximated sample still a meaningful and reasonable counterpart sample in this case? Since LNA approximates the samples/representations by inquiring about the neighbors, I think it can't make a good estimation for unseen data. Although GAN-based approaches don't have such limitations, I admit the LNA is a low-cost and easy way to approximate mirror samples.
> >
> >
> > - In the last, I appreciate the authors conduct extra experiments to compare with more baselines. I would suggest filling out Table 1 by extending their code to new datasets. Since most of works provide their code, it is not hard to replace the dataset and carry out some results. You can also check their paper carefully, as I know, SHOT has the result for office-home, the average accuracy is 71.8%

---

> > > ### Author Response · Authors · 2021-08-21
> > > **Hope our discussion can help to make the idea well-understood**
> > >
> > > We would like to thank the reviewer for the further feedback. We're happy to discuss about our work, which is enjoyable and inspiring.
> > >
> > > 1. Thanks for pointing out the missing superscripts. The first term would be $KL(q^s(\tilde {x}^s(x_j^t))||q^t(x_j^t))$. This equation actually minimizes the discrepancy of the two relative positions wrt to their own domains. Sorry again for the confusion and we'll fix it in the next version.
> > >
> > > 2. Discussion about the extreme case of "toy"example
> > >
> > >    We like this example and we can clarify our methods with CyCADA from the following two aspects.
> > >
> > >    -  Feasibility:
> > >
> > >    Empirically, if we use the proposed method in this case, the anchors for one domain would be the average representation 'Daily calendar' while in the other domain would be average representation 'Weekly calendar'.
> > > The LNA cannot generate ''daily calendar'' using ''weekly calendar'', but it would try the best to select the most similar samples to construct mirror samples.(i.e. the most similar "daily calendar" in many patterns such as "the grid around the number" except the "weekly layout" pattern).
> > > Then ER assures the "relative" distances to the anchors("weekly calendar" center & "daily calendar" center) to be equivalent, i.e. the deviations (e.g. "the grid around the number") of the mirror samples from the average "daily calendar" are similar to the mirror sample wrt the "weekly calendar" (In fact, other class centers can also regularize the relative equivalence, c.f. Eq 4 & 5).
> > > In this way, the mirror loss actually aligns the "relative roles" of the equivalent sample in their own domains, i.e. the covariate shift.
> > > CycADA might generate "daily calendar" samples to solve this issue, while the proposed method aligns relative equivalence conditioning on the alignment of the centers of "daily calendar" and "weekly calendar".
> > >
> > >     - Limitation:
> > >
> > >    However from the point of ideal alignment, our proposed method has limitation in this case since 1) the anchors (class centers) are not ideally equivalent cross domains, 2) the LNA cannot generate ideal mirrors.
> > > We kindly remind the reviewer to see line 335-337 in our main paper. We have clarified the limitation of the proposed method if the samples cross domains would be highly scarce and biased.
> > > This ''toy example'' falls into this range. Our proposed method assumes the domain gap is the covariate shift and the samples are not too much biased.
> > > The results Table 4 also shows large improvement, but still cannot reduce the error rates to an absolute low level.
> > > That's why we use 'reducing' instead of 'eliminating' in the title.
> > >
> > >
> > >    We expect our claimed mirror sample, which has theoretic support in performance improvement, can attract the community's further research.
> > > For the extreme case like this example, combining CyCADA with Mirror would be a solution to reduce the high biasedness(i.e. generating the weekly calendar for the other domain and then use LNA based on both the generated sample as well as real sample). But it would involve the generation models and loose the conciseness compactness.
> > > We will add those discussion, combined with the initial responses to the final version.
> > >
> > >
> > > 3. Thanks for the suggestions. We actually have run many recent sota methods on those dataset.
> > > We do not present those results in order to avoid the confusion about "reported results'' and "reproduced results''.
> > > We'll fill those gaps with the results with different footnote.
> > > An example of the reproduced results for CAN & SHOT would be as follows
> > >
> > > |  |Office-31  | Office-Home |  CLEF |   VisDA
> > > --|       --|         --|          -- | --|
> > > CAN |   90.6  |    68.8* |   89.6* |   87.2
> > > SHOT |  88.7    | 71.6    | 87.2*   | 79.6*+
> > > Ours |  91.1 |  72.7 |   91.6 |   87.9
> > >
> > > \* means reproduced result while the others are reported results in original papers.
> > >
> > > In sum, we would thank the reviewer's kindly discussion again. We expect our clarifications can further improve the evaluation about our work.

---

> > > > ### Comment · Reviewer_ZzRU · 2021-08-23
> > > > **updated the score**
> > > >
> > > > Thanks for the detailed reply. The clarifications were helpful and I have accordingly increased my score from 4 to 6. Looking forward to read the revised version.

---

### Official Review · Reviewer_2iWh · 2021-07-08

**Rating:** 6
**Confidence:** 4

**Summary:**

This paper is addressing the domain adaptation problem as previous methods neglect the structural properties of the underlying distribution. This paper proposes a virtual mirror concept with Local Neighbor Approximation (LNA) and Equivalence Regularization (ER). The proposed method achieves state-of-the-art performance on several benchmarks.

**Limitations And Societal Impact:**

The potential negative societal impact is not discussed in this paper.

**Main Review:**

1. claim concerns:
a) For Equivalence Regularization (ER), this paper adopts clustering to obtain the class centers for target domains. However, it is critical to have reliable clusters as the anchors. How to make sure this would happen? Moreover, if the target clusters are not reliable, this will also distort the internal structure of the intrinsic underlying distribution, which is one of the main claims in the paper. The details about this concern are not clear in this paper.

2. technical detail concerns:
a) Equation (7) consists of two parts: the relation between target samples and source virtual mirrors (first loss term), and the relation between source samples and target virtual mirrors (second loss term). It would be great to discuss more these two terms. For example, which term is more critical in this problem? Why?

3. experiment concerns:
a) There are some other state-of-the-art papers missing in this paper. For example, RSDA (CVPR’ 20) also achieves competitive performance on several benchmarks (e.g., 91.1 on Office-31, 70.9 on Office-Home, and 90.5 on ImageCLEF). FixBi (CVPR’ 21) adopts similar ideas that generate virtual samples and calculate the consistency between source and target domains. Moreover, FixBi also achieves state-of-the-art performance (e.g., 91.4 on Office-31, 72.7 on Office-Home, and 87.2 on VisDA-2017). It would be better to include them in the experiment comparison (also compare the difference of approaches).

References:
1. (RSDA) Xiang Gu, et al. "Spherical Space Domain Adaptation with Robust Pseudo-label Loss", CVPR, 2020.
2. (FixBi) Jaemin Na, et al. "FixBi: Bridging Domain Spaces for Unsupervised Domain Adaptation", CVPR, 2021.

b) The proposed method contains two parts: Local Neighbor Approximation (LNA) and Equivalence Regularization (ER). The ablation studies only investigate the details about ER (in Table 2). The contribution of LNA is not very clear. It would be better to investigate LNA as well. For example, designing experiments to adopt LNA but with other types of loss functions (e.g., those in SRDC [41]).


**Time Spent Reviewing:**

6

---

> ### Author Response · Authors · 2021-08-10
> **Responses to Reviewer 2iWh**
>
> We thank the reviewers' suggestions. The concerns are insightful and push us to review our work more comprehensively.
> We hope the following responses can relive the concerns.
>
>
> #### 1. Concerns about ''Reliable clusters''
> It is true that reliable clusters for target domain play an important role ER.
> To enhance the reliability of the target clusters, the following 2 strategies are used.
> 1)Using the k-means cluster with the source class centers as the initialized centers (line 223-228) rather than the pseudo-label by the model trained in source data.
> Since the K-Means cluster has stability guarantees (an Expectation-Maximation scheme), it can provide the local optimal assignment for the class considering reliable source data centers(they're based on labels).
> 2) In implementation, we further combine both source center and target center as the real centers in ER stated in line 225-226, which can alleviate the unreliability.
> We have to admitted that the reliable pseudo-label generation still lacks theory understanding in community. It's effects in transfer learning, semi-supervised learning are future work.
>
> #### 2. concerns about ''Two loss terms impacts''
> Thanks to point out this topic. Actually we have evaluated those impacts during investigation. The experiments on Office-home are listed as follows. In the experiment,  we only apply the ER terms on backbone FC, with each KL loss on s $\to$ t and t $\to$ s separately.
> Although there are differences for different tasks, averagely, using either one of them has relatively little impacts on the final results.
> Considering the symmetry states of the loss, we provide the two terms together.
> We did not present these results in the original submission considering the brittle impacts of this factor. We'll add the following analysis in the next version to make reader more clear.
>
> |Method  |  Ar-Cl| Ar-Pr | Rw-Cl | Rw-Pr | Pr-Ar | Pr-Rw |  avg|
>
> |s $\to$ t  | 52.1 |   74.1 |  59.0 | 84.8 |  70.1 |  81.8 |   71.7|
>
> |t $\to$ s   |51.9 |   75.3 |  58.1 | 84.4 |  70.7 |  81.9 |   71.5|
>
>
> #### 3. About the ''missing citations''
> Thanks to remind the missing sota results, we will definitely add those results in the final version.
> To be specific, the ''FixBi'' uses the idea of mix-up to generate two intermediate domains for large domain gap. Although it involves generating virtual samples, it is not designed for the dilemma that we pointed out in Sec2.
> ''FixBi'' utilizes both source and target samples while our methods only use one domain samples.
> In experiments, our results still have competitive results comparing on several benchmarks.
> We will add those analysis in the final version to make the experiment more thoroughly sound.
>
> #### 4. Ablation about ''LNA''
> To evaluate the variations of LNA, we evaluate the different weighting scheme in Table 9 of Appendix.
> The different distance metrics has brittle impacts on the results.
> For the different combinations of ''LNA'' with other losses, we want to claim the current ''LNA'' are designed working with ''ER'' and cannot be isolated.
> As stated in Line 130-139 of main paper, ''LNA'' only uses local neighbors and cannot assure the generated samples are the counterpart.
> That's why we introduce the ER, which regularizes the generated samples cross domains.
> SRDC's loss only applies in one domain and cannot satisfy our model purpose.
> We hope the explanation can relieve the reviewer's concerns.

---

> > ### Comment · Reviewer_2iWh · 2021-08-18
> > **Thank you for the rebuttal**
> >
> > The authors address all my concerns in the rebuttal. However, I agree with Reviewer ZzRU that this paper should compare with other domain adaptation methods that also have similar ideas with generated virtual samples (CyCADA is only one of the examples). Without this comparison, the contribution of the proposed paper will be less obvious.

---

> > > ### Author Response · Authors · 2021-08-21
> > > **Thank you for the feedback**
> > >
> > > Thanks for the feedback and we're happy to see the responses have addressed the reviewer's concerns.
> > > We actually analyze CyCADA methods in the related work part in line 314-318, but don't compare with them experimentally since those methods have little reported results on traiditional DA benchmarks like Office-31, Office-home.
> > > They mainly work on pixel-level task for domains with style difference.
> > >
> > > We have provided the analysis in the response to Reviewer ZzRU.
> > > In sum, the generation-based methods have relative strong assumption about domain gaps, i.e. the style difference, while our method does not assume the shift form.
> > > We also compare the performances with the CyCADA in the response.
> > > Those analysis will be added into our paper to make our contribution more distinguishable.
> > > Thanks for the comments and please do not hesitate to post messages if you have further concerns.

---

> > > > ### Comment · Reviewer_2iWh · 2021-08-26
> > > > **Thank you for the response**
> > > >
> > > > I read the authors' discussion with Reviewer ZzRU. Looking forward to reading the revised version.

---

### Official Review · Reviewer_SJwM · 2021-07-17

**Rating:** 6
**Confidence:** 5

**Summary:**

To reduce domain shift (covariate shift due to different domain distributions), the authors propose a loss based on Equivalence Regularization and Mirrored Samples.

The main idea is to: 1) find class prototype on both source and target domain and 2) encourage each sample's relative embeddings in the source and target domains to be similar.

Experiments are carried out on 4 datasets and obtained competitive accuracy against SOTA methods.


**Ethical Concerns:**

There is no ethical concerns

**Limitations And Societal Impact:**

Yes.

**Main Review:**

Strengths:
- The idea of using mirrored samples to align domain embedding is quite interesting.
- The author proposes to ER (Equivalence Regularization) to compute domain anchors as class prototypes for each domain.
- Relative vector coordinates for each sample can then be computed relative to domain anchors.
- Evaluation and ablations are conducted on 4 datasets, showing competitive accuracy against SOTA methods.

Weaknesses:
- The LNA and ER are run per-epoch, I am not convinced how they can effectively alter the embedding, if the datasets are large.
- The author seems to assume both source and target have groundtruth labels. Is this a common practice in the domain adaptation usecases? Usually the target domain label is not accessible.

**Time Spent Reviewing:**

7

---

> ### Author Response · Authors · 2021-08-10
> **Responses to Reviewer SJwM**
>
> We thank the reviewers' comments and suggestions. We're happy to see our idea is interesting.
> What's more, this interesting idea is motivated by the optimal transport theory(part A of supplementary materials) , whose asymptotic properties are analytically analyzed(Sec.4 of the main paper and Part C of supplementary materials). We expect the mirror idea can motivate more insights in distribution alignment cross domains.
> We hope the following responses can relieve the reviewer's concerns and further improve the evaluations.
>
> #### 1. "Embeddings evolvement using LNA and ER per-epoch"
> LNA and ER utilize the embeddings (centers) per-epoch, which are similar in calculating the centers in SRDC[1], TPN[2], CAN[3], etc.
> We investigate the embedding evolvement during training and show them in Fig.6 and Fig.2 of the appendix for Office-Home, which is a relative large dataset.
> By zooming up, we can see that the cluster shape of both source and target domains are approaching similar with the training going on.
> Even for VisDA, the effectiveness of embeddings evolvement can be validated by the final accuracy improvement, e.g. for the per-epoch update methods like CAN, we outperform in AP by 0.7, and by 2-3% in the challenging tasks (Table4 of appendix).
>
> #### 2. Concerns about "accessing the target label".
> The whole method follows the traditional UDA settings, i.e. no accessing the target label.
> As stated in line 174-178 of main paper and the Algorithm1 in Appendix, we use k-means to produce target class center initialized by the source class centers.(Strictly speaking, we even did not use the sample-level pseudo label.)
> The ER only utilizes the relative distance to those centers.
> To further reduce the noise for the centers, we also combine the centers of both source and target (line 223-224).
>
> [1] H Tang,et al. Unsupervised domain adaptation via structurally regularized deep clustering, CVPR20.
>
> [2] Y Pan et al, Transferrable Prototypical Networks for Unsupervised Domain Adaptation, CVPR19 .
>
> [3] G Kang,et,al. Contrastive adaptation network for unsupervised domain adaptation, CVPR19 .

---

### Decision · Program_Chairs · 2021-09-27

**Decision:**

Accept (Poster)

**Comment:**

The paper proposes a method for unsupervised domain adaptation (UDA) by constructing "virtual mirror" samples of source domain in target domain (and vice versa) and then "aligning" the corresponding mirror pairs across domains for matching feature distributions. The method is novel (reviewer zzRU raised a concern on limited novelty by pointing out the connection with CyCADA but I think the proposed idea is sufficiently different) and shows strong empirical performance. The method has some heuristic components as pointed out by reviewers (eg, reliance on k-means to get clusters, the method to construct mirrors) however the authors have pointed out connections to optimal transport theory in their response. I suggest the authors make this more prominent in the revised version of the paper. Overall, the paper is above the acceptance threshold in my view.